# Exploring the Association between Socioeconomic Environment, Eating Habits and Level of Nutrition in Children of High School Age: A Part of National Survey

**DOI:** 10.3390/children11091074

**Published:** 2024-08-31

**Authors:** Mladen Grujicic, Marija Sekulic, Jovana Radovanovic, Viktor Selakovic, Jagoda Gavrilovic, Vladan Markovic, Marija Sorak, Marko Spasic, Rada Vucic, Snezana Sretenovic, Olivera Radmanovic, Zlata Rajkovic Pavlovic, Katarina Nikic Djuricic, Jovica Tomovic, Milena Maricic, Snezana Radovanovic

**Affiliations:** 1Department of Hygiene and Epidemiology, Health Centar Bijeljina, 76300 Bijeljina, Republic of Srpska, Bosnia and Herzegovina; mgrujicic@gmail.com; 2Department of Hygiene and Ecology, Faculty of Medical Sciences, University of Kragujevac, 34000 Kragujevac, Serbia; 3Department of Epidemiology, Faculty of Medical Sciences, University of Kragujevac, 34000 Kragujevac, Serbia; 4Department of Communication Skills, Ethics and Psychology, Faculty of Medical Sciences, University of Kragujevac, 34000 Kragujevac, Serbia; 5Department of Infectious Diseases, Faculty of Medical Sciences, University of Kragujevac, 34000 Kragujevac, Serbia; 6Department of Radiology, Faculty of Medical Sciences, University of Kragujevac, 34000 Kragujevac, Serbia; 7Department of Gynecology and Obstetrics, Faculty of Medical Sciences, University of Kragujevac, 34000 Kragujevac, Serbia; 8Department of Surgery, Faculty of Medical Sciences, University of Kragujevac, 34000 Kragujevac, Serbia; drmspasic@gmail.com; 9Department of Internal Medicine, Faculty of Medical Sciences, University of Kragujevac, 34000 Kragujevac, Serbia; 10Clinic for Rheumatology and Allergology, University Clinical Center Kragujevac, 34000 Kragujevac, Serbia; 11Faculty of Medical Sciences, University of Kragujevac, 34000 Kragujevac, Serbia; 12Department of Dentistry, Faculty of Medical Sciences, University of Kragujevac, 34000 Kragujevac, Serbia; 13Department of Psychiatry, Faculty of Medical Sciences, University of Kragujevac, 34000 Kragujevac, Serbia; kaca_nikic@msn.com; 14Department of Pharmacy, Faculty of Medical Sciences, University of Kragujevac, 34000 Kragujevac, Serbia; 15Academy of Applied Studies Belgrade, The College of Health Sciences, 11070 Belgrade, Serbia; 16Department of Social Medicine, Faculty of Medical Sciences, University of Kragujevac, 34000 Kragujevac, Serbia; jovanarad@yahoo.com

**Keywords:** obesity, children of high school age, demographic and socio-economic characteristics, national health survey, Serbia

## Abstract

Background/Objectives: One of the raising public health problems in the adolescent population is obesity, which has reached epidemic proportions worldwide. The aim of this work is to determine the prevalence of obesity in the population of children of secondary school, age 15 to 19 years in Serbia and the determinate connection with demographic and socio-economic characteristics of the respondents and their eating habits and physical activity. Methods: The research is part of the fourth National Population Health Survey conducted in 2019, which was conducted by the Republic Institute of Statistics, in cooperation with the Institute of Public Health of Serbia and the Ministry of Health of the Republic of Serbia. As a research instrument, questionnaires were used in accordance with the methodology of the European Health Survey. For the purposes of this research, data on the adult population aged 15–19 and over were used. Results: The association of overweight with demographic and socioeconomic characteristics was examined using binary regression. In the univariate model, male gender (OR = 1.95), younger age (OR = 1.57) and Region of Vojvodina (OR = 2.47) stood out as significant predictors of overweight, which was confirmed by the multivariate model. Conclusions: The results of our study emphasize that the prevalence of obesity in the population of high school youth is at a significant level and that a lot more needs to be done to promote healthy lifestyles and raise awareness of their benefits on health status.

## 1. Introduction

One of the raising public health problems in the adolescent population is obesity, which has reached epidemic proportions worldwide, where the occurrence of severe obesity raised roughly four times in more than three decade. Most obese youth continues to sustain excessive adiposity level in later life, endangering them to complications that obesity carry along such as numerous chronic non-communicable diseases and mental health issues which negatively affect their social and emotional health [1].

The prevalence of obesity among children and adolescents, according to the available data in the period from 2020 to 2035 is predicted to increase rapidly, with an increase of 5% to 33% in different regions, depending on the region and gender. Also, among children and adolescents in low-income countries, there is likely to be a dramatic increase in the prevalence of obesity during this period, from 4% to 13% in girls, and from 2% to 6% in boys, in middle-income economies of 5% up to 14% in girls and from 6% to 16% in boys, which is significantly less compared to children and adolescents in countries with higher middle incomes that will experience a very high level of obesity prevalence with 31% in girls and 40% in boys [2,3,4].

Serbian population health surveys between 2006 and 2013 also document an increase in the prevalence of overweight and obesity among children and adolescents in Serbia from 2.6% to 4.9% [5].

Changes of eating habits, physical inactivity, and a sedentary lifestyle are cited as the most significant factors contributing to the increase in the obesity epidemic [6,7,8].

For this reason, starting from the fact that a good knowledge of the health behaviors of children of secondary school age in an environment enables the formation of strategies for health promotion and prevention of obesity, the aim of this work is to determine the prevalence of obesity in the population of children of secondary school age 15 to 19 years in Serbia and determinate the relationship with demographic and socio-economic characteristics of the respondents and their eating habits and physical activity.

## 2. Materials and Methods

### 2.1. Type of Study

The research was conducted as a cross-sectional study on the territory of the Republic of Serbia. It is the fourth national population health survey of Serbia, in the period from 5 October to 30 December 2019, which was implemented by the Ministry of Health of the Republic of Serbia in cooperation with the Institute for Public Health of Serbia “Dr. Milan Jovanović Batut” and the Republic Institute of Statistics. The research was conducted in accordance with the methodology and instruments of the European Health Survey-third wave, EHIS-wave 3 [9].

### 2.2. Target Population

The primary target population consisted of persons aged 15–19 years. Persons who live in households and institutions (student and school dormitories, homes for children and youth with disabilities, homes for socially vulnerable children), and persons who were not provided consent were excluded from research.

### 2.3. Sample Size and Sample Allocation

This research used stratified two-stage sample which was carried out according to the place of living (other and urban settlements) and four regions devided by the geographical areas (Belgrade region, Vojvodina region, Sumadija and Western Serbia region, South and Eastern Serbia region). The population census conducted in the Republic of Serbia in 2011 was used as a framework for sample selection. A sample of 5.114 households was realized, in which a total of 15.621 persons were registered, of which 13.589 persons were aged 15 and over. For the purposes of this research, a total of 739 respondents, both genders, aged 15 to 19 were examined.

### 2.4. Ethical and Legal Aspects

Ethical standards are in line with the international Declaration of Helsinki, as well as the legislation of the Republic of Serbia. Signed informed consent for participation in the research was obtained from each respondent.

### 2.5. Research Instruments

The research instrument were standardized questionnaires constructed in accordance with the European Health Interview Survey (EHIS—European Health Interview Survey, wave3) which were then adapted to the specifics of our area. Following questionnaires were used:The household info panel, for the the socio-economic characteristics of the household members;A questionnaire for each member’s age 15 years and aboveA questionnaire for self-completion, was used because of the sensitivity of the questions considering topics of alcohol and drug abuse, smoking or sexual behavior, which were not suitable for the face-to-face method.Form for the measurement of anthropometric values such as height, weight and blood pressure.

The Physical Activity Questionnaire—European Health Interview Survey (EHIS-PAQ-European Health Interview Survey-Physical Activity Questionnaire) was used to assess physical activity [9].

An electronic, decimal, scale for medical purpose was used to measure body mass of students. An adjustable SECA altimeter was used to measure body height.

In children and adolescents, the Body Mass Index (BMI) was calculated in the same way as in adults, but the interpretation of the obtained values is different. For this reason, calculation of BMI by the CDC for children 2–19 years was used which calculated and grouped each children by the percentile values for a specific gender and age of the child. According to the CDC guidelines, the underweight category had a value of <5 percentile, normal weight 5–85 percentile, pre-obese 85–95 percentile and obese ≥95 percentile [10]. 

### 2.6. Variables

The independent variables used in the research are: sociodemographic (gender, age, family structure, region, well-being index) and health determinants (eating habits, physical activity), while the dependent variable was the level of nutrition (BMI).

### 2.7. Statistical Methods

The Statistical Package for Social Sciences software (SPSS Inc, version 20.0, Chicago, IL, USA) was used for statistical analysis. To compare differences between groups two tests were performed, chi-square for analyzing non parametric data and ANOVA for parametric data. Afterwards, the relationship between dependent variables and a series of independent variables was analyzed by bivariate and multivariate logistic regression. The risk was assessed using the OR (odds ratio) size, with a 95% confidence interval. Statstically significant were the results with *p* value less than 0.05 (*p* < 0.05). 

## 3. Results

The research was conducted on a sample of 739 high school students aged 15–19, of which 56% were boys and 44% were girls. Demographic and socioeconomic characteristics are shown in Table 1.

Eating habits in the examined population of high school students looked like this: 89.5% students had regular breakfast and consumed fruits in 33.8% 1–3 times a week, vegetables were consumed 4–6 times a week in 74.4% and 1–3 times per week 33.8% consumed pure juice. 

Physical activity in the examined population of high school students looked like this: 25.1% of students walked less than 150 min per week, 69% of them used bikes less than 150 min per week but half of them (50.6%) had recreation in duration more than 150 min per week. In free time more than a half of them (58.5%) were never physically active and 82.5% of them never used exercise to strengthen their muscles. Nearly half of them (45.7%) spent 4–7 h in a day in sedentary position. 

The analysis of the distribution of respondents according to nutrition categories showed that in the examined population two thirds of respondents (74.8%) had normal nutrition, 14.2% were pre-obese, 7.3% were obese and 3.6% were underweight.

The relationship between sociodemographic characteristics and aware on nutrition is shown in Table 2. Determinants that stood out as a significant with nutrition levels were gender, age groups and years of high school students. A significantly lowerpercentage of female respondents (12.1%) were in the pre-obese category compared to males (16.0%), while 10.3% of male and 3.5% of female high school students were obese. The ANOVA test showed that the differences in mean values of age between nutrition categories are statistically significant (*p* = 0.003). The average age of the respondents who belong to the category of obese is 16.8 ± 1.4, while the average years of age in normal weight was 17.3 ± 1.3.

Observed by age group, younger high school students are pre-obese (16.4%) and obese (8.8%) in a higher percentage than older high school students. With the age of the respondents, the participation of pre-obese and obese high school students decreases. Although there is no correlation between the degree of obesity and the region, high school students from the region of Vojvodina are more likely to be pre-obese (20%), while the obese most often come from the region of Southern and Eastern Serbia (9.9%).

In relation to eating habits, only consumption of vegetables or salad, excluding potatoes and juices made from fresh vegetables stood out as a statistically significant variable related to the respondents’ nutrition levels (*p* = 0.012). 36.6% of pre-obese respondents and 42.9% of obese respondents eat vegetables or salad once or more times per day, while smaller percentage of underweight respondents (33.3%) eats vegetables or salad once or more times per day. There were no significant differences between different nutritional categories and other investigated variables. Also, there was no statistically significant difference between physical activity and nutritional level in the studied population (Table 3).

The association of overweight with demographic and socioeconomic characteristics was examined using binary regression. In the univariate model, male gender (OR = 1.95), younger age (OR = 1.57) and Region of Vojvodina (OR = 2.47) stood out as significant predictors of overweight. The multivariate regression model singled out male gender (OR = 1.85), younger age (OR = 1.47) and Region of Vojvodina (OR = 1.78) as the most significant predictors of overweight (Table 4).

## 4. Discussion

Data from the Health Behavior Survey of School-Age Children covering 36 countries in the WHO European Region shows that the prevalence of overweight/obesity in high school students ranges from 5% to more than 25% in some countries. A comparison of data from various studies showed that the situation is not improving and that global prevalence of obesity among children keeps rising [11]. 

More developed countries shows interesting pattern of change of the obesity, from younger adults to the younger population. Obesity in the children age is becoming noticeably in the focus of public health, and accurate determination of a cohort of children and young adults on an annual basis is a key requirement for developing and implementing interventions to reduce obesity [12]. 

Our results showed that 14.2% of the examined population of high school students was pre-obese, while 7.3% of the respondents were obese. Broken down by gender, a higher percentage of pre-obese (16%) and obese (10.3%) boys are present.

Similar results to ours were recorded in the Nordic countries, where 49% of male and 37% of female adolescents were overweight. About 14% of Norwegian adolescents were obese [13]. 

The prevalence of obesity and overweight among young people is often strongly correlated with certain dietary habits, such as low fruit and vegetable intake, skipping breakfast or widely available fast food and its frequent consumption. When it comes to behavioral patterns related to eating habits and physical activity among Finnish adolescents, it was observed that there is a decrease in the daily intake of fresh vegetables, especially among young men, which may indicate negative eating habits and may be associated with an increase in obesity rates [14]. 

Results from United States, surveyed among youth, showed that among high school students nationwide, roughly every fifth ate fruit or consumed pure fruit juice three or more times a day, while vegetable consummation was even lesser—only 15.3% [15]. Our results showed the following distribution in the regularity of consumption of certain food groups: fruits and vegetables are consumed once or more a day by 34.4% and 18.3% of high school students, juices from pure fruits less often than once a week (33.3%).

The results found in a study conducted in Spain showed that the daily intake of vegetables and fruits was 50.8% versus 36% [16]. 

Other studies have also observed lesser fruit and vegetable consumption among adolescences, and age is considered a determining factor. The trend of decreasing consumption of fruits and vegetables begins in childhood and continues into youth [17]. 

When it comes to eating habits, regular and proper meals are very important to ensure adequate growth and development. Children of high school age are often picky when choosing food, and in addition, they often skip meals due to obligations and suffer from an unbalanced diet. Many studies show that school children often skip breakfast and that high school students who skip breakfast are more likely to be obese [18]. Our results are not completely consistent with the findings of other studies, because the high school students in our study show almost 90% regularity of eating breakfast every day. 

Although our study population of high school students is not physically active, there was no correlation between physical activity and the level of nutrition in our study population.

Similar results in terms of physical activity were recorded in research in the United States of America, where about 50% of high school students reported that they did not engage in physical activity with duration of 60 min in 5 days a week in the last week compared to the survey [19]. 

Although there are many well-known health benefits that comes along with consistent physical activity, there are no notable increase in performing aerobe and anaerobe physical activities among adolescents which are proven to be beneficial not only for better memory and brain function, physical health and vitality but also, those activities are helpful as the stress and depression reducer [20]. It is worrisome that children from different income countries are adapting to the “new fashion” of increased sedentary lifestyle and less physical activity [21]. 

Physical activity levels have also been noted to decline as children progress from childhood to adolescence [22]. Similar results to ours were observed in China where nearly every fifth student engaged in any type of daily physical activity lasting 60 min or more, while 85.8% engaged in sedentary behavior for more than 2 h a day, and only 15–34% met recommended guidelines for physical activity [23]. 

A study that aimed to examine the relationship between age and the prevalence of healthy behavior among adolescents from 37 countries and regions showed that in boys, daily fruit and vegetable consumption is represented by 16.2% and daily physical activity by 18.9%. Among girls, daily physical activity was represented by 9.4%, and daily fruit and vegetable consumption by 22.6% at the age of 15 [24].

Many studies show that the socioeconomic status of the family influences the occurrence of obesity [12], which was also recorded in our research, where high school students who belong to the poorest strata according to the welfare index are pre-obese (15.7%) and obese (8%), which is probably the result of insufficient financial resources to choose healthier foods. 

Similarly children with lower household income show lower levels of physical activity and increased sedentary time than children from higher household income families [25]. 

When it comes to family structure, in families with children where both parents are present, the children were most often normally well-nourished (7.3%), while in incomplete families, children were more often obese (11.8%), which is probably due to the fact that one parent is more busy in order to acquire enough material resources and therefore does not have enough time to follow the diet and the selection of food, and on the other hand, in dysfunctional families there is usually insufficient money, which is documented by many studies [26]. This is supported by our data, where high school students belonging to the poorest strata are pre-obese (15.7%) and obese (8.0%) in a higher percentage.

There was noticed repetitive pattern of behavior among adolescents living in the European countries with smaller income levels to those who live in higher income Western countries that reflects in different aspects such as more common sedentary lifestyle with followed lesser physical activity, increase of unhealthy habits like sugar consumption and fast food and increased screen time which all potentially lead to the overweight or obesity [27].

The situation in developed countries of Western Europe and North America showed that increasing prevalence of obesity and overweight in girls and higher incidence in boys are strongly correlated with poor physical activity and bad eating habits [28]. 

### Strengths and Limitations

Our study presents some limitations. The main one is the correlation between the socioeconomic environment, eating habits and level of nutrition should be verified in longitudinal studies. Also, the lack of this research is a more comprehensive coverage of other factors that can contribute to the development of obesity and are important in order to form healthy habits and monitor the long-term outcome. Although there are areas for future research, we believe this study provides valuable evidence because it included a vulnerable category, and these are children whose number in this research is significant in order to determine which factors lead to obesity and its consequences in this vulnerability population. Also, this research will contribute significantly to the development of evidence-based public health strategies with broader applicability and impact.

## 5. Conclusions

The results of our study emphasize that the prevalence of obesity in the population of high school youth is at a significant level and that much more needs to be done to promote healthy lifestyles and raise awareness of their potential benefits for health status. Determinants that stood out as a significant with nutrition levels were gender, age groups and years of high school students. In relation to eating habits, only consumption of vegetables or salad, stood out as a statistically significant variable related to the respondents’ nutrition levels. In the univariate model, male gender, younger age, and Region of Vojvodina stood out as significant predictors of overweight which was confirmed by multivariate regression model. Therefore, health education interventions are needed to ensure an increase in healthy lifestyles and consequently reduce the prevalence of obesity in this vulnerable population.

## Figures and Tables

**Table 1 children-11-01074-t001:** Sociodemographic characteristics of the respondents.

Variable	Total		Gender			
			Male		Female	
	*n*	%	*n*	%	*n*	%
**Age (years)**						
15	123	16.6	63	15.8	60	17.6
16	152	20.6	85	21.3	67	19.2
17	148	20.0	72	18.0	76	22.4
18	177	24.0	98	24.6	79	23.2
19	139	18.8	81	20.3	58	17.1
**Age groups**						
15–17	398	53.9	209	52.4	189	55.6
18–19	341	46.1	190	47.6	151	44.4
**Region**						
Belgrade region	149	20.2	73	18.3	76	22.4
Region of Vojvodina	170	23.0	93	23.3	77	22.6
The region of Sumadija and Western Serbia	256	34.6	139	34.8	117	34.4
Region of Southern and Eastern Serbia	164	22.2	94	23.6	70	20.6
**Type of settlement**						
Urban settlements	420	56.8	233	58.4	187	55.0
Village settlements	319	43.2	166	41.6	153	45.0

**Table 2 children-11-01074-t002:** Socio-demographic characteristics and nutritional status.

Variables	Nutritional Levels								*p* *
	Underweight		Normal Weight		Pre-Obese		Obesity		
	*n*	%	*n*	%	*n*	%	*n*	%	
**Gender**									
Male	10	3.1	225	70.5	51	16.0	33	10.3	**0.005**
Female	11	4.3	206	80.2	31	12.1	9	3.5	
**Mean age ± SD**	17.7 ± 1.4		17.3 ± 1.3		16.8 ± 1.4		16.8 ± 1.4		**** 0.003**
**Age (years)**									
15	3	4.1	47	64.4	14	19.2	9	12.3	**0.007**
16	1	0.8	86	67.7	29	22.8	11	8.7	
17	3	2.6	97	82.9	9	7.7	8	6.8	
18	7	4.9	111	77.6	17	11.9	8	5.6	
19	7	6.0	90	77.6	13	11.2	6	5.2	
**Age groups**									
15–17	7	2.2	230	72.6	52	16.4	28	8.8	**0.028**
18–19	14	5.4	201	77.6	30	11.6	14	5.4	
**Region**									
Belgrade region	6	7.0	69	80.2	8	9.3	3	3.5	0.120
Region of Vojvodina	5	4.2	84	70.0	24	20	7	5.8	
The region of Sumadija and Western Serbia	4	1.8	173	75.9	33	14.5	18	7.9	
Region of Southern and Eastern Serbia	6	4.2	105	73.9	17	2.0	14	9.9	
**Household type**									
One parent with children	2	5.9	25	73.5	3	8.8	4	11.8	0.557
Both parents with children	10	4.1	187	77.3	31	12.8	14	5.8	
Other households	9	3.0	219	73.0	48	16.0	24	8.0	

* the significant results are mention, where they are less than 0.05, ** This results was done by ANOVA for parametric data, the other where nonparametric. So we wanted to make clear which test was performed for that variable.

**Table 3 children-11-01074-t003:** Eating habits and physical activity of students and nutritional status.

	Underweight		Normal Weight		Pre-Obese		Obesity		
	*n*	%	*n*	%	*n*	%	*n*	%	*p*
**Breakfast regularity**									
Every day	17	81.0	387	89.8	75	91.5	38	90.5	0.172
Sometimes	4	19.0	39	9.0	7	8.5	2	4.8	
Never	0	0.0	5	1.2	0	0.0	2	4.8	
**Fruit, excluding juices made by squeezing fresh fruit or fruit concentrate**									
Once or more times a day	8	34.8	158	36.7	22	26.8	15	36.6	0.270
4–6 times a week	8	34.8	145	33.6	24	29.3	9	22	
1–3 times a week	6	26.1	113	26.2	30	36.6	12	29.3	
Less than once a week	1	4.3	14	3.2	5	6.1	5	12.2	
Never	0	0.0	1	0.2	1	1.2	0	0.0	
**Vegetables or salad, excluding potatoes and juices made from fresh vegetables**									
Once or more times a day	7	33.3	205	47.6	30	36.6	18	42.9	**0.012**
4–6 times a week	10	47.6	131	30.4	23	28.0	8	19.0	
1–3 times a week	4	19.0	78	18.1	28	34.1	11	26.2	
Less than once a week	0	0	11	2.6	1	1.2	4	9.5	
Never	0	0	6	1.4	0	0.0	1	2.4	
**Juices made from 100% pure fruit or vegetables. excluding juices made from concentrate**									
Once or more times a day	0	0	23	5.3	6	7.3	0	0	0.383
4–6 times a week	1	4.3	40	9.3	7	8.5	6	14.6	
1–3 times a week	7	30.4	126	29.2	19	23.2	13	31.7	
Less than once a week	9	39.1	159	36.9	29	35.4	12	29.3	
Never	6	26.1	81	18.8	21	25.6	10	24.4	
**Walking (min/week)**									
Less than 150 min	6	3.8	115	73.2	27	17.2	9	5.7	0.561
More than 150 min	15	3.6	311	75.1	55	13.3	33	8	
**Bicycling (min/week)**									
Less than 150 min	5	2.9	18	68.4	25	14.5	14	8.1	0.956
More than 150 min	2	2.4	59	72	13	15.9	8	9.8	
**Physical activity in free time**									
Never	12	3.6	246	73	50	14.8	29	8.6	0.316
1 to 2 days a week	5	6.5	59	76.6	7	9.1	6	7.8	
3 to 7 days a week	4	2.5	126	78.3	24	14.9	7	4.3	
**Sports/fitness/recreation minutes/week**									
Less than 150 min	6	5.7	81	77.1	13	12.4	5	4.8	0.456
More than 150 min	2	1.8	86	77.5	17	15.3	6	5.4	
**Muscle strengthening exercises**									
Never	18	3.8	360	76.1	62	13.1	33	7	0.443
1 to 2 days a week	1	1.5	44	66.7	14	21.2	7	10.6	
**Time spent in a sitting position**									
Up to 3 h	6	3.3	132	72.5	25	13.7	19	10.4	0.660
4 to 7 h	9	4	169	74.4	35	15.4	14	6.2	
8 or more hours	4	2.8	110	76.9	21	14.7	8	5.6	

**Table 4 children-11-01074-t004:** Cross-over ratios (ORs) and 95% confidence intervals (CIs) for the association of demographic and socioeconomic characteristics of subjects with overweight.

Variables	Univariate Model		Multivariate Model	
	OR (95%CI)	*p*	OR (95%CI)	*p*
**Gender**				
Female	1		1	
Male	1.95 (0.96–3.95)	**0.002**	1.85 (0.76–3.25)	**0.013**
**Age groups**				
15–17	1.57(1.04–2.36)	**0.030**	1.47 (1.15–2.27)	**0.042**
18–19	1		1	
**Region**				
Belgrade region	1		1	
Region of Vojvodina	2.47 (1.17–5.25)	**0.018**	1.78 (0.63-.96)	**0.019**
The region of Sumadija and Western Serbia	1.95 (0.96–3.95)	0.063	1.33 (1.09–1.61)	0.354
Region of Southern and Eastern Serbia	1.98 (0.94–4.18)	0.072	1.25 (0.82–1.94)	0.754
**Index of well-being**				
Rich layer	1		1	
Middle layer	0.91 (0.48–1.73)	0.79	1.14 (0.98–1.32)	0.651
Poor	1.24 (0.77–1.99)	0.371	1.18 (0.97–1.39)	0.462
**Type of settlement**				
Urban settlements	1.09 (0.72–1.65)	0.67	1.29 (0.74–1.84)	0.770
Village settlements	1		1	

## Data Availability

Data are unavailable due to privacy or ethical restrictions because the current owner of the rights, the Institute of Public Health of Serbia, “Milan Jovanović Batut” and the database was handed over to the University of Kragujevac with an official letter for the purpose of further research.

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
