# Peer review of "Exploring the Association between Socioeconomic Environment, Eating Habits and Level of Nutrition in Children of High School Age: A Part of National Survey"

_children, 2024, doi:10.3390/children11091074_

Round 1
Reviewer 1 Report
Comments and Suggestions for Authors
The article presented is mostly well-written. Below are my comments and suggestions:
· Chapter 2.3. I suggest including a study outline along with an indication of the number of participants included in the study and excluded from participation in the study along with the reason.
Additionally, in my opinion, this chapter should focus only on the age of the people who are being studied within this article (from 15 to 19 years old).
· Chapter 2.5. The questionnaires used should be included in the supplementary materials.
· Table 1. In my opinion the line "total" placed at the very end is misleading and should be placed after each variable: age, region,…..
· Should the Index of well-being be included in the table on socioeconomic characteristics (table 1 and table 2)?
· In summarizing the discussion, it would be beneficial to highlight the strengths and weaknesses of the study.
Author Response
Dear Reviewer,
Thank you for your thoughtful comments and constructive feedback on our manuscript. We appreciate the time and effort you took to review our work. Your insights have been invaluable in improving the quality and rigor of our research. Below, we have addressed each of your comments and explained the modifications made to the manuscript accordingly.
- Chapter 2.3. I suggest including a study outline along with an indication of the number of participants included in the study and excluded from participation in the study along with the reason. Additionally, in my opinion, this chapter should focus only on the age of the people who are being studied within this article (from 15 to 19 years old).
Response: Dear, thank you for your suggestion, we keep focus on children (from 15 to 19 years old), and we removed unnecessary data on other age groups like you said.
- Chapter 2.5. The questionnaires used should be included in the supplementary materials.
Response: Questionnaires used in this research are unavailable due to privacy because the current owner of the rights, the Institute of Public Health of Serbia, “Milan Jovanović Batut” and the database was handed over to the University of Kragujevac with an official letter for the purpose of further research.
- Table 1. In my opinion the line "total" placed at the very end is misleading and should be placed after each variable: age, region,…..
Response: Adding additional row for each variable in Table 1 will end up with the same total number and percentages as in the row total, at the very end. So there will be just additional rows for the same numbers, for each variable.
In the Table 2, the total is describing the percentage portions of respondents with different levels of nutrition. In this case, the number of respondents belonging to the each level of nutrition will remain same for each variable. It would be more confusing adding the additional same numbers.
Both “total” rows are not necessary for results so we will omit them from both tables, as the reviewer found it misleading, but we explained in previous paragraphs the reasons of why we put them at the first place.
- Should the Index of well-being be included in the table on socioeconomic characteristics (table 1 and table 2)?
Response: We have removed index of well-being from the tables as it is not the sociodemographic characteristic. Thank you for this observation, it was mistakenly added.
- In summarizing the discussion, it would be beneficial to highlight the strengths and weaknesses of the study.
Response: According to your recommendations we wrote in summarizing the discussion, the strengths and limitations of the study. Our study presents some limitations. The main one is the correlation between the socioeconomic environment , eating habits and level of nutrition should be verified in longitudinal studies. Also, the lack of this research is a more comprehensive coverage of other factors that can contribute to the development of obesity and are important in order to form healthy habits and monitor the long-term outcome. Although there are areas for future research, we believe this study provides valuable evidence because it included a vulnerable category, and these are childrena whose number in this research is significant in order to determine which factors lead to obesity and its consequences in this vulnerability population. Also, these research will contribute significantly to the development of evidence-based public health strategies with broader applicability and impact.
Reviewer 2 Report
Comments and Suggestions for Authors
It is a timely study and provide many positives and negatives.
Following areas can be added to improve the quality of the paper.
Line 57-58: Severe obesity - Is it obesity or anything else, if not better to change or provide reference
Line 94: add 15-19 years
Line 95: What is collective HH? Please define
Line 97: Better to change to people who were not provided consent; otherwise it is necessary to explain ethical part.
Line 138: Was calculated
Line 140: Were used; please check all and change to past tense
BMI calculation and CDC and WHO application. It is needed to know which indices that you used. usually 15-19 years, BMI categorisation is not valid. Need to interpret using age and sex specific BMI according to WHO. Please verify what reference that you used. Need to mention why both, it is not provided in the results section.
Line 176: Need to provide the principle of calculating well-being, may be in the method section. New Nutrition - Better to change to aware on nutrition, however, it is not provided in the results section.
Line 306-310: Rephrase considering the significant findings of this study rather than providing general recommendations.
Author Response
Dear Reviewer,
Thank you for your thoughtful comments and constructive feedback on our manuscript. We appreciate the time and effort you took to review our work. Your insights have been invaluable in improving the quality and rigor of our research. Below, we have addressed each of your comments and explained the modifications made to the manuscript accordingly.
Comments: It is a timely study and provide many positives and negatives.
Following areas can be added to improve the quality of the paper.
- Line 57-58: Severe obesity - Is it obesity or anything else, if not better to change or provide reference
Response: Dear, according to your recommendations, we corrected severe obesity in obesity.
- Line 94: add 15-19 years
Response: We added 15-19 years.
- Line 95: What is collective HH? Please define
Response: We erase collective , and we kept households and institutions.
- Line 97: Better to change to people who were not provided consent; otherwise it is necessary to explain ethical part.
Response: We change into: The primary target population consisted of persons aged 15-19 years. Persons who live in collective households and institutions (student and school dormitories, homes for children and youth with disabilities, homes for socially vulnerable children), persons who do not know how to read and write, and persons who were not provided consent were excluded from research. persons who were not provided consent.
- Line 138: Was calculated
Response: We change in was calculated.
- Line 140: Were used; please check all and change to past tense
BMI calculation and CDC and WHO application. It is needed to know which indices that you used. usually 15-19 years, BMI categorisation is not valid. Need to interpret using age and sex specific BMI according to WHO. Please verify what reference that you used. Need to mention why both, it is not provided in the results section.
Response: We checked all and changed it to past tense.
Response: In children and adolescents, the Body Mass Index (BMI) was calculated in the same way as in adults, but the interpretation of the obtained values is different. For this reason, calculation of BMI by the CDC for children 2-19 years was used which calculated and grouped each children by the percentile values for a specific gender and age of the child.According to the CDC guidelines, the underweight category had a value of <5 percentile, normal weight 5-85 percentile, pre-obese 85-95 percentile and obese ≥ 95 percentile. We put reference for this part, number 10. Centers for Disease Control and Prevention. Child and Teen BMI Categories. Available from:http://www.cdc.gov/bmi/child-teen-calculator/bmi-categories.html
- Line 176: Need to provide the principle of calculating well-being, may be in the method section.
Response: Dear, the index of well-being is mistakenly added to the tables where we only compared our variables with sociodemographic characteristics. As the index of well-being is socioeconomic characteristics we have removed that variable from tables.
- New Nutrition - Better to change to aware on nutrition, however, it is not provided in the results section.
Response: It is changed.
- Line 306-310: Rephrase considering the significant findings of this study rather than providing general recommendations.
Response: This paragraph was added in conclusion. Determinants that stood out as a significant with nutrition levels were gender, age groups and years of high school students. In relation to eating habits, only consumption of vegetables or salad, stood out as a statistically significant variable related to the respondents' nutrition levels. In the univariate model, male gender, younger age, and Region of Vojvodina stood out as significant predictors of overweight which was confirmed by multivariate regression model.